# The Effectiveness of Water- versus Land-Based Exercise on Specific Measures of Physical Fitness in Healthy Older Adults: An Integrative Review

**DOI:** 10.3390/healthcare12020221

**Published:** 2024-01-16

**Authors:** Athanasios A. Dalamitros, Eirini Toupektsi, Panagiota Alexiou, Stamatia Nousiou, Vicente Javier Clemente-Suarez, José Francisco Tornero-Aguilera, George Tsalis

**Affiliations:** 1Laboratory of Evaluation of Human Biological Performance, School of Physical Education and Sport Sciences, Aristotle University of Thessaloniki, 54124 Thessaloniki, Greece; toupekts@phed.auth.gr (E.T.); panalexiou5@gmail.com (P.A.); stamnous@phyed.duth.gr (S.N.); tsalisg@phed-sr.auth.gr (G.T.); 2Faculty of Sports Sciences, European University of Madrid, 28670 Madrid, Spain; vicentejavier.clemente@universidadeuropea.es (V.J.C.-S.); josefrancisco.tornero@universidadeuropea.es (J.F.T.-A.); 3Grupo de Investigación en Cultura, Educación y Sociedad, Universidad de la Costa, Barranquilla 080002, Colombia

**Keywords:** aquatic exercise, exercise on land, fitness parameters, older individuals, review

## Abstract

As the population ages, maintaining an active lifestyle becomes increasingly vital to promote overall health and well-being in older individuals. Water- and land-based exercises have emerged as popular options, each offering a distinct set of benefits tailored to the unique needs of this population group. An electronic database search, including PubMed, Scopus, MEDLINE, and Web of Science, was conducted until 15 September 2023, using the Preferred Reporting Items for Systematic Reviews and Meta-Analyses (PRISMA) guidelines, to investigate the effects of water-based compared to land-based exercise on selected fitness parameters in older healthy individuals. The eligibility criteria included studies with at least two groups of participants aged 60 and older, with physical fitness outcome measures. A total of ten studies met the inclusion criteria and were analyzed. While both exercise modalities may offer significant benefits, this review’s findings emphasized the absence of conclusive evidence and consensus for recommending a single exercise category applicable to aquatic or land environments and providing more definite guidance to improve health-related physical fitness parameters in healthy older individuals. Finally, combining both training approaches may lead to a comprehensive array of health benefits for this age group population by also considering individual’s needs, preferences, and fitness goals.

## 1. Introduction

As part of the natural aging process, a wide spectrum of physiological and pathological changes in the population unfolds [1]. Maintaining a healthy lifestyle and well-being in older individuals becomes more imperative. Participation in sports and physical activities stands as a cornerstone of this endeavor as an effective and easy way [2], presenting a plethora of physical and mental health benefits. In this direction, an increasing body of research and systematic reviews has well documented the effectiveness of land-based exercises (LBE) (i.e., involving activities conducted on solid ground) in improving physical fitness measures [3] including improvements in muscle strength [4], cardiorespiratory fitness [5], bone health [6], balance [7], flexibility [8], body composition [9], and overall quality of life [10] in older adults.

The type of the selected exercise is a critical factor in maximizing the potential health-related effects of this age group population [11], offering distinct benefits according to an individual’s goals and physical condition state. In this sense, water-based (WBE), or aquatic, exercises (conducted in the buoyant embrace of pools and aquatic environments) have gained increasing popularity among older individuals and healthcare professionals as a “gentler” approach, especially for those with an elevated risk of injury [12], due to issues related to the warmth of the water [13] and the decreased mechanical loads on skeletal joints and soft tissues of the lower extremities [14]. Moreover, this type of exercise is associated with increased energy expenditure [15] and reduced likelihood of acute injury [16].

Given the above, the choice between LBE and WBE programs becomes a pivotal decision requiring careful consideration of older adults’ unique needs, demands, and goals. In the current scientific literature, there are randomized controlled trials dedicated to comparing the effects of LBE versus WBE programs on older individuals with a variety of chronic health conditions, osteoporosis [17], knee osteoarthritis [18,19,20], hypertension [21], and low back pain [22], or in patients after a coronary event [23]. To the best of our knowledge, no review has thoroughly examined relevant information in healthy older populations. Therefore, the purpose of this integrative review was to analyze research findings related to the effects of LBE versus WBE programs on health status and specific physical fitness measures in this cohort of subjects. This information is essential for tailoring exercise programs and optimizing health outcomes by empowering healthcare professionals to make informed choices, ensuring that exercise remains a safe and effective tool for promoting health and longevity. 

## 2. Materials and Methods

### 2.1. Study Design

This study involves an integrative review methodology, aiming to analyze and summarize the modifications in physical fitness parameters in healthy older adults who underwent an exercise protocol on land versus in aquatic environments. The search strategy was performed on four electronic databases (PubMed, Scopus, MEDLINE, and Web of Science) using the Preferred Reporting Items for Systematic Reviews and Meta-Analyses (PRISMA) guidelines. A comprehensive synthesis of search terms was applied, using the following keywords and their synonyms: “water-based exercise OR aquatic exercise OR in-water exercise OR aquatic activity” AND “land-based exercise OR on-land exercise OR land-based activity” AND “physical fitness OR fitness outcomes OR physical fitness parameters” AND “healthy older adults OR older individuals”. The initial search yielded 232 studies. Then, 67 duplicates were removed, and 126 studies were excluded after reviewing their titles and abstracts. The full texts of the remaining 10 studies were selected and analyzed. Figure 1 demonstrates a detailed flow chart of the respective process.

### 2.2. Study Selection

The eligibility criteria included studies with training interventions with at least two study groups of participants who received either an LBE or a WBE program, with no chronic disease, aged 60 or older, on physical fitness outcome measures, until 25 September 2023, and only written in the English language. Moreover, there was no restriction regarding publication period. Studies that did not include an evaluation of physical fitness outcomes (cardiorespiratory fitness, muscle strength, flexibility, and body composition) or not published in peer-reviewed journals were excluded. 

### 2.3. Data Extraction

Relevant information was extracted from each study by two of the authors (E.T. and P.A.) after reviewing the title and the abstract of each study. Quality checking was conducted accordingly to check the full text of the articles. Following this, the authors independently decided the eligibility of each study for inclusion. No disagreements between the two authors were reported in this part. The extracted data included the author, publication year, study design, participant characteristics, intervention types, study characteristics, physiological outcomes, measures, and study findings. 

### 2.4. Quality Assessment

The authors chose to develop their quality assessment checklist after pilot testing including a subset of studies. Nine different criteria were applied, each one evaluated in every selected study. Those with positive scores > 5 (assigned with the mark “+”) were characterized as “high quality”, while in a different case, the study was described as “low quality”. Table 1 summarizes the selected studies’ criteria for quality assessment.

## 3. Results

### 3.1. Characteristics of the Included Studies 

Our results identified 10 eligible studies with a total of 693 participants, ranging from 60 to 75 years old. Four studies were classified as “high-quality” [25,26,31,32]. Eight of them employed randomized controlled trial study design. The participants performed either an LBE or a WBE program (n = 9). In one study [28], the effects of a WBE program versus a combination of LBE and WBE were investigated. The exercise programs had an intervention length varying from 10 weeks to 6 months, with a weekly frequency of one to five days per week, and a duration of 30 to 90 min per training session was applied. In one study [28], the effects of five years of regular engagement in exercise programs executed on land or in water were examined. The intensity at which the exercise programs were performed was set according to the participant’s heart rate [24,25,28,31,32] or the score based on self-reported measures (Borg scale) [26,27,29,33]. Four studies included a control group [25,29,31,32]. The selected studies were conducted in Australia (n = 2), Canada (n = 1), Italy (n = 1), Spain (n = 1), South Korea (n = 2), Japan (n = 1), Brazil (n = 1), and Iran (n = 1). 

### 3.2. Main Outcomes

#### 3.2.1. Cardiorespiratory Fitness

Four studies included measures of cardiorespiratory fitness, assessed during direct (oxygen consumption measurement) [24,25,31,32] or indirect methods (walk test) [30]. Three of them were classified as high-quality studies [25,31,32]. Haynes et al. [32] reported similar benefits of two protocols of walking exercise on VO_2_max values, maximal heart rate values, and time to exhaustion. Likewise, Taunton et al. [24] observed analogous improvements in VO_2_peak values. On the contrary, the study of Bocalini et al. [25] concluded that there was a more significant improvement in VO_2_max values in the WBE group (42 vs. 32%). Finally, in the study of Askari et al. [30], the WBE group experienced more profound increases (42.2 vs. 21.4%). Hence, different evidence regarding the effectiveness of land- versus water-based exercise programs to improve cardiorespiratory fitness is realized. 

#### 3.2.2. Muscle Strength

Among the 10 selected studies, 7 included muscle strength measurements of the upper and lower body by using different tests, such as the hand-grip test (n = 3), the arm-curl test (n = 2), and the chair-stand muscle endurance test (n = 3). The curl-up and the push-up tests were applied in one study [24], while in three cases, dynamometers were used to evaluate muscle strength of the hip and the knee joint [27,28,33]. The retrospective study of Tsujimoto et al. [28] analyzed data from a large number of clinical examinations and compared the long-term effects (five-year follow-up) of participating in either a water-based program or a combination of land-based and in-water exercise, unsupervised, programs. No differences between the groups were observed, as both exercise modalities failed to maintain muscle strength values (hand-grip strength). Two high-quality studies [25,26] reported mixed results. In the Bergamin et al. study [26], a statistically significant increase in hand-grip strength (26.1%) was only noticed in the LBE group. In addition, Bocalini et al. [25] showed larger increases in the upper limbs’ strength values during the exercise in the water, while the LBE group only increased lower limb strength values, but to a lesser extent, compared to the WBE group. In comparison, calf muscle strength was increased in the WBE group (5.5 and 14.6%, for the isotonic and isometric tests, respectively). Moreover, both groups maintained their lower body strength values six months after the intervention period. In contrast, Oh and Lee [33] reported more profound increases in hip, knee, and hand-grip measures in the WBE group, during different testing points, whereas comparable improvements were shown during the chair-stand and the arm-curl test. The WBE group of Oh et al.’s study [27] reported significantly lower body muscle strength improvements (hip abduction and adduction), but no differences between the two groups were observed for the hip flexion and extension values. Both exercise groups in the Padua et al. study [29] managed to enhance leg extension strength values to the same degree, while abdominal muscles showed slightly larger increases in the WBE group (15 vs. 16%). Finally, Taunton et al. [24] found improvements during a 60 s curl-up test (abdominal endurance) only in the LBE group. Therefore, it appears the selected studies yield inconclusive results regarding the effectiveness of a single method over the other for improving muscle strength of the upper and lower limbs in healthy older adults.

#### 3.2.3. Flexibility

Flexibility was considered in seven studies. Of those, one study solely applied the fingertip-to-floor test [29], while in one case, the sit and reach test was the only one applied [24]. The rest of the selected studies measured both upper- and lower-part flexibility by using the back scratch along with the sit and reach test. Taunton et al. [24], as well as Oh et al. [27], reported no significant improvement in flexibility values after the execution of either the WBE or LBE programs. Opposite results were shown in three studies [25,29,30]; that is, significant improvements were found in both groups. Finally, in the high-quality studies of Bocalini et al. [25] and Bergamin et al. [26], flexibility values of the upper body were only increased after implementing the WBE programs, whilst lower body flexibility was improved in both groups. In summary, as with the above-mentioned measures of physical fitness (cardiorespiratory fitness and muscular strength), the inconclusive nature of the results reported emphasizes the lack of strong evidence pointing to one method as superior to enhancing flexibility in healthy older populations. 

#### 3.2.4. Body Composition

Only two high-quality studies assessed body composition. A significant reduction, focused on measures of total mass (4%) and trunk fat mass (5.3%), was reported in the study of Bergamin et al. [26] only in the WBE group. In contrast, the intervention program applied in the study of Naylor et al. [31] concluded positive effects in both groups, as evaluated using central adiposity measures. As such, the available evidence related to the effectiveness of WBE versus LBE interventions on body composition values in healthy older adults is limited, and no definite conclusion can be drawn. Here, it should be noted that in three studies [24,28,32], the body mass index was evaluated, showing no significant modifications in both groups. Nevertheless, this index does not provide any information related to the distribution of body fat or body mass whatsoever [34]. Table 2 describes each study’s characteristics, the physiological outcomes measured, and the findings of the included studies.

## 4. Discussion

To the authors’ knowledge, this is the first review examining the effects of WBE versus LBE intervention programs on a variety of physical fitness parameters, and more precisely cardiorespiratory endurance, muscle strength, flexibility, and body composition, in healthy older individuals. It should be noted that dynamic balance, a leading parameter affecting fall risks [35], was not included in the current study, as a recent systematic review and meta-analysis [36] offered a comprehensive understanding of the topic. According to the data analyzed, the existing scientific literature does not provide sufficient evidence in favor of one exercise modality over the other (water- versus land-based exercise intervention). This lack of consensus can be partially explained by the heterogeneity in study designs, the variations in participants’ characteristics, as well as the diversity regarding the physical fitness parameters assessed. 

When considering the effects of WBE or LBE alone on measures of physical fitness, in the respective age group analyzed here, different conclusions have been reported. For instance, the systematic review by Bergamin et al. [16] that analyzed the effectiveness of water-based exercise interventions in older adults reported strong evidence underpinning the use of this type of exercise to promote cardiorespiratory fitness and muscle strength. Similar conclusions were reported in a more recent systematic review [37] that underlined the positive responses on functional fitness after participating in WBE programs in healthy older adults. In this case, specific recommendations regarding the training characteristics (i.e., frequency, duration, intensity) were also provided. In contrast, the review study of van der Bij et al. [38], including home- or group-based exercise interventions, reported trivial changes, characterized as short-lived. On the other hand, the systematic review and meta-analysis by Yang et al. (2019) [39] showed that when aiming to enhance physical fitness in older healthy adults through intervention including land aerobic and resistance training, 30 to 40 min of exercise, executed with a frequency of two to three sessions per week, continued for 12 to 16 weeks, is required. 

In the case of analyzing the effectiveness of WBE or LBE programs in terms of physical fitness measures, various factors should be taken into consideration. During the data analysis, different study designs were observed regarding the execution of cardiorespiratory fitness protocols. For example, the Haynes et al. study [32] included two protocols of walking exercises, executed either on land or in water, at similar intensities, with a progressive exercise duration (40 to 65% of heart rate reserve), during a 24-hour period. On the other hand, in the Taunton et al. study [24], the participants followed a 12-week aerobic exercise program at a constant intensity corresponding to 60 to 65% of the individuals’ maximal heart rate, of only 20 min in length. In the case of Bocalini et al. [25], the water-based endurance-type program lasted for 45 min, at an intensity corresponding to 70% of the age-predicted maximum heart rate with an implementation period of 12 weeks. Finally, in the Askari et al. study [30], the participants followed a 15-minute aerobic WBE program for a total of six weeks, while no analytical information regarding the exercise intensity was provided. On the contrary, analogous methodology inconsistencies were less apparent for the measure of muscle strength. For instance, the majority of the intervention programs in the included studies (five out of seven studies) had an approximate duration of 40 minutes, involving the same muscle groups, while the intensity was set between 65 and 70% of each participant’s age-predicted maximal HR [25] or evaluated by using the Borg scale rating on an equal intensity (3 to 4 on the Borg’s Scale from 0 to 10) [29] without progression during the intervention period. 

The diversity during the measure of muscle strength among the included studies should also be displayed here. For instance, isokinetic dynamometry (applied in three studies) consists of a controlled and highly precise measure [40,41], while functional tests such as those applied in the selected studies (e.g., push-ups, sit-ups, and arm curls) assess this parameter in a more practical context. Thus, focusing on different muscle groups or specific movements may result in discrepancies in the results and make it difficult to draw meaningful conclusions. In addition, testing conditions such as the time of the day, a critical factor to be considered during muscle strength assessment [42], were not apparent in all studies. On the contrary, when evaluating cardiorespiratory fitness and flexibility, a greater methodological coherence was observed, as in almost all the included studies 3 min stages of continuous exercise performed on a treadmill and the sit-and-reach test, respectively, were used.

Only in three cases [24,25,33] a comprehensive analysis of the physical fitness profile was performed. In the rest of the studies, the measure of cardiorespiratory fitness was absent, while two high-quality studies focused on a single parameter, namely aerobic endurance [32] and body composition [31]. Moreover, an unequal gender ratio among participants was observed. In three cases [27,31,33], the gender of the participants was not mentioned. Four studies included both men and women [26,28,29,32], two only involved women [24,25], while in the study of Askari [30], the participants were solely men.

Although not established, muscle strength is presented to be more trainable in older male individuals [43]. On the contrary, although different mechanisms are involved, comparable increases in VO_2_max values (19 vs. 22% for male and female participants, respectively) after a moderate endurance-type training program have been reported [44]. In addition, the recent review and meta-analysis performed by Markov et al. [45] demonstrated that older male and female individuals (aged from 50 to 73 years old) respond differently to a concurrent strength and endurance training period. In their study, they reported larger, but no significant, effects on measures of muscle strength and cardiorespiratory endurance in female participants. Moreover, the authors highlighted that caution should be taken when presenting gender-related differences in intervention studies, as the absolute or relative expression of the results may impact the data presented. 

Along the same line, when assessing muscle strength values in men and women, several limitations and considerations should be considered by researchers and healthcare professionals. These, among others, include the different muscle mass, muscle function, and hormonal profiles [46,47], as well as the lack of including a representative sample of both genders to avoid selection bias. Further concerns to discuss include the absence of information related to any adverse effects of the exercise programs implemented in the selected studies. Furthermore, among the 10 studies included in this review, 3 had a duration of 6 to 10 weeks [27,30,33]. According to a previous review on older individuals, cardiorespiratory endurance and muscle strength responses are greater after training periods of more than 12 weeks [48]. Finally, it should be mentioned that body composition was the least-examined measure of physical fitness. This fact follows the previous review by Bergamin et al. [16] examining the effectiveness of water-based programs in physical fitness parameters of the respective age group. Body composition has been related to the gradual loss of muscle strength, which can lead to implications on functional performance [49]. Hence, future studies considering this parameter are needed. 

An obvious limitation of the current systematic review is that the authors developed and used a custom quality assessment checklist. This method was selected as the specific characteristics of the studies reviewed here did not align with those proposed, for example, by the Cochrane Collaboration or the Newcastle–Ottawa scale. For instance, the similarities between the different groups (WBE or LBE) or the number of dropouts, two of the criteria applied in the Cochrane Collaboration tool, were hard to notice in the reviewed studies. Moreover, the use of a post hoc analysis in the selected studies (one of the criteria considered here) can potentially offer a practical significance of any differences between the groups observed. Consequently, a decision was made to include criteria based on different sections of the selected studies, namely methods, statistical analysis, and limitations. In any case, the subjectivity of the assessment tool applied here was minimized as two experts in the field reviewed the entire procedure, ensuring its validity and reliability. Another limitation is the language bias, as only studies written in English were selected. As a result, studies published in other languages may contain relevant information that was missed. Finally, we acknowledge that a mix of high- and low-quality studies and only a small number of studies were included; therefore, part of the findings presented here might be potentially not reliable. 

## 5. Conclusions

To summarize, from the available literature and the criteria set in this review, there is currently no conclusive evidence for recommending one specific category of exercise that takes place either in water or on land when focused on improving health-related physical fitness parameters in healthy older adults. Future studies involving healthy older individuals of both genders, with higher methodological quality and a long-term design while covering a wider range of standardized and validated physical fitness measures, are encouraged. Moreover, exploring the dose–response relationship of exercise characteristics (i.e., intensity, duration, frequency) more extensively, and extending research beyond physical fitness by also investigating underlying biological mechanisms and the impact on life quality, can offer crucial information for the overall well-being of this age group population. When referring to clinical practice and policy making, emphasizing consistency, safety, and progress monitoring is of foremost importance. Finally, integrating both training approaches may lead to a wider range of benefits for this age group population. Importantly, considering the individual’s preferences, needs, and enjoyment, while, in parallel, consulting healthcare professionals is important before participating in any new exercise routine. 

## Figures and Tables

**Figure 1 healthcare-12-00221-f001:**
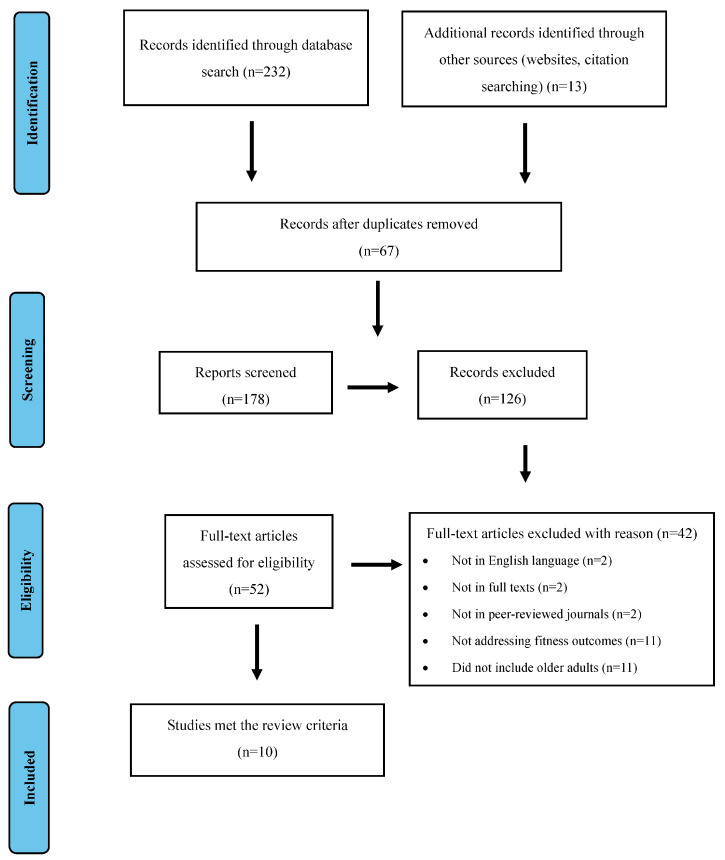
Preferred Reporting Items for Systematic Reviews and Meta-Analyses flow chart of each stage of the study selection.

**Table 1 healthcare-12-00221-t001:** Quality assessment of the selected studies.

Study	Randomized	Inclusion/Exclusion Criteria	Matched Intensity	Control Group	Supervised Programs	Multiple Assessment Times	Follow-Up	Post Hoc Analysis	Limitations	Results
Taunton et al. (1996) [24]	-	+	+	-	+	-	-	+	+	5/9
Bocalini et al. (2008) [25]	+	+	+	+	+	-	-	+	+	7/9
Bergamin et al. (2013) [26]	+	+	+	-	+	-	-	+	+	6/9
Oh et al. (2014) [27]	+	+	+	-	+	-	-	-	+	5/9
Tsujimoto et al. (2017) [28]	-	+	-	-	-	-	-	-	+	2/9
Padua et al. (2017) [29]	+	+	+	+	+	-	-	-	-	5/9
Askari et al. (2018) [30]	+	+	-	-	+	-	-	-	+	4/9
Naylor et al. (2020) [31]	+	+	+	+	+	+	-	-	+	7/9
Haynes et al. (2019) [32]	+	+	+	+	+	-	+	-	+	7/9
Oh & Lee (2021) [33]	+	+	-	-	-	-	+	-	+	4/9

**Table 2 healthcare-12-00221-t002:** Studies’ characteristics, physiological outcomes, and findings.

Study	Participants/Methods/Evaluation	Measures/Instrumentation/Testing Procedures	Outcomes
Taunton et al. (1996) [24]	• 41 F (70 ± 3.2 years) • Duration: 12 weeks • Intervention: • WBE (n = 23): training in water (45 min/ day, 3 days/week, 60–65% HR_max_) • LBE (n = 18): training on land (45 min/ day, 3 days/week, 60–65% HRmax) • Evaluation: T_1_: pre-intervention T_2_: at 6 weeks (halfway) T_3_: post-intervention	Anthropometric data: • BMI • Sum of skinfolds • Waist/hip ratio Muscular strength and endurance: • Grip strength • Curl-ups • Push-ups	• No significant differences in anthropometric data for any group between the 3 evaluation points. • Significant ↑ in number of curl-ups was found for LBE from T_1_ to T_3_. • No significant differences in the grip strength and the number of push-ups were found for any group between the 3 evaluation time points.
		Flexibility: • Trunk forward flexion (sit-and-reach test)	• No significant differences in the trunk forward flexion for any group between the 3 evaluation points.
		Cardiorespiratory fitness • VO_2_ peak (modified Balke treadmill protocol monitored by Beckman Metabolic measurement cart)	• Significant ↑ in VO_2_peak for both groups from T_1_ to T_3_. No significant differences between the 2 groups.
Bocalini et al. (2008) [25]	• 50 sedentary F (62–65 years) • Duration: 12 weeks • Intervention: • WBE (n = 25): training in water level at or near xiphoid level (60 min/day, 3 days/week, @70% HRmax) • LBE (n = 15): walking (60 min/day, 5 days/week, @70% HRmax) • Control group: (CG; n = 10) • Evaluation: 1st: pre-intervention 2nd: post-intervention	• Body weight	• No significant differences in body weight values for any group.
		• HRrest	• Significant ↓ in HRrest for WBE.
		Cardiorespiratory fitness: • VO_2_max (Bruce treadmill protocol monitored by 12-lead electrocardiography)	• Significant ↑ in VO_2_max for both groups. Significantly higher values of the WBE compared to LBE.
		Neuromuscular parameters: • Upper body strength (arm-curl test) • Lower body strength (chair-stand test) • Agility (8-foot up-and-go test) • Lower body flexibility (sit-and-reach test) • Upper body flexibility (back-scratch test)	• Significant ↑ in upper body strength and upper body flexibility for WBE. • Significant ↑ in lower body strength, agility, and lower body flexibility for the WBE & LBE. Significantly higher values of the lower body flexibility of WBE compared to LBE.
Bergamin et al. (2013) [26]	• 59 (29 M/30 F; 71.2 ± 3.2 years) • Duration: 24 weeks • Intervention: • WBE (n = 20): training in hot spring water (36 °C) (60 min/day, 2 days/week, 13–16 grade on Borg’s RPE scale) • LBE (n = 20): training on land (60 min/day, 2 days/week, 13–16 grade on Borg’s RPE scale) • Control group: (CG; n = 19) • Evaluation: T_0_: pre-intervention T_1_: post-intervention	• Body mass (BWB—800 AS scale)	• Significant ↓ in the body mass, the body mass index, the trunk fat mass, the total fat mass, and the radius fat area for the WBE. Significant ↑ in tibia muscle density for WBE. • Significant ↑ in the trunk fat-free mass, total fat-free mass, the radius muscle density, and the tibia total area for LBE. • Significant ↑ in the total fat mass and the tabia total area was found for CG. Significant ↓ in trunk fat-free mass and total fat-free mass for CG.
		Physical performance and strength: • Upper body flexibility (back-scratch test) • Lower body flexibility (sit-and-reach test) • Grip strength (hand grip dynamometry) • Isotonic [KET] and isometric [KEM] knee extension strength (dynamometric load cell applied to a knee extension device)	• Significant ↑ in lower body flexibility for the WBE and LBE. Significantly higher values of the lower body flexibility of WBE compared to LBE. • Significant ↑ in upper body flexibility for WBE. • Significant ↑ in grip strength for LBE & CG. • Significant ↓ in KET and KEM for CG.
Oh et al. (2014) [27]	• 66 (>65 years) • Duration: 10 weeks • Intervention: • WBE (n = 34): (60 min/day, 3 days/week) • LBE (n = 32): (60 min/day, 3 days/week) • Evaluation: 1st: pre-intervention 2nd: post-intervention	Lower body muscle strength (handheld dynamometer): • Hip flexion • Hip extension • Hip abduction • Hip adduction • Flexibility (back-scratch test, chair sit-and-reach test)	• Significant ↑ in hip abduction and adduction strength was found for both groups. Significantly different values between the 2 groups. • No significant differences of flexibility values between the 2 groups.
Tsujimoto et al. (2017) [28]	• 77 subjects (10 M/67 F ≥ 60 years) • Duration: 5 years • Intervention: • WBE group (n = 38): (30–60 min/day, at least 1 day/week) • Combination of WBE & LBE (n = 39): WBE & LBE for 30–60 min/day, at least 1 day/week each • Evaluation: annual	BMI	• No significant differences of the BMI and the systolic BP for any group.
		Functional parameters: • Grip strength (Smedley’s hand dynamometer)	• Significant ↓ in grip strength for both groups.
Padua et al. (2017) [29]	• 132 subjects (49 M/83 F, 68.9 ± 5.5 years) • Duration: 8 months • Intervention: • WBE (n = 44): training in water (50 min/ day, 2 days/week, 3–4 grade on Borg’s RPE scale) • LBE (n = 44): training in the gym (50 min/day, 2 days/week, 3–4 grade on Borg’s RPE scale) • Control group (CG; n = 44) • Evaluation: Baseline: pre-intervention Follow–up: post-intervention	Physical skills: • Hamstring flexibility (fingertip-to-floor test) • Leg extensor strength (chair-stand test) • Abdominal muscle strength (reverse-crunch test)	• Significant ↑ in all the physical skills for both groups. Significantly higher abdominal strength values in WBE compared to LBE.
Askari et al. (2018) [30]	• 60 M (>60 years) • Duration: 6 weeks • Intervention: • WBE (n = 30): (60 min/ day, 3 days/week) • LBE (n = 30): (60 min/day, 3 days/week) • Evaluation: 1st: pre-intervention 2nd: post-intervention	Biomotor abilities: •Flexibility (Wells test) • General endurance (6MWT)	• Significant ↑ in all the biomotor abilities for both groups. Significantly higher general endurance values in WBE compared to LBE.
Naylor et al. (2019) [31]	• 71 subjects (62.5 ± 6.8 years) •Duration: 24 weeks/48 weeks (60 subjects) • Intervention: • WBE (n = 23): water walking (60 min/day, 3 days/week, 40–65% HRR) • LBE (n = 23): land walking (60 min/day, 3 days/week, 40–65% HRR) • Control group • C (n = 25) • Evaluation: baseline/24 weeks/48 weeks	Body composition parameters	• Significant ↓ in the central adiposity for both groups from baseline to week 24. • Significant ↑ in lower limb lean for WBE from baseline to week 24.
Haynes et al. (2019) [32]	• 71 (19 M/52 F, 62.1–62.7 years) • Duration: 24 weeks • Intervention: • WBE (n = 25): water walking (150 min/ week, 3 days/week, 45–65% HRR) • LBE (n = 13) land walking (150 min/week, 3 days/week, 45–65% HRR) • Control group (CG; n = 23) • Evaluation: baseline 24 wk.: immediately post-intervention 48 wk.: 24 weeks post-intervention	Cardiorespiratory fitness: • VO_2_max (electrochemistry oxygen analyzer and carbon dioxide analyzer) • VO_2_ • Exercise duration • HRmax (12-lead electrocardiogram)	• Significant ↑ in VO_2_max for WBE and LBE groups from baseline to week 24. • Significant ↑ in time to exhaustion for LBE. • No significant differences of any values for any group from baseline to week 48.
Oh & Lee (2021) [33]	• 66 (4 M/62 F > 65 years) • Duration: 10 weeks • Intervention: • WBE (n = 34): (60 min/day, 3 days/week) • LBE (n = 32): (60 min/day, 3 days/week) • Evaluation: T_1_: pre-intervention T_2_: post-intervention T_3_: 1-year post-intervention	Senior fitness tests: • Chair stand test • Arm curl test • Two-minute step test • Chair sit-and-reach test • Back scratch test • Timed up-and-go test	• Significant ↑ in all the senior fitness tests from T_1_ to T_2_ and from T_1_ to T_3_ for both groups. Significance ↑ in the chair sit-and-reach test, the back-scratch test, and the timed up-and-go test from T_2_ to T_3_ for both groups.
		Hip strength (handheld dynamometer): • Flexion • Extension • Abduction • Adduction • Interval rotation • External rotation	• Significance ↑ in all the hip strength parameters from T_1_ to T_2_ and from T_1_ to T_3_ for both groups. Significant differences in hip flexion, extension, adduction, and external rotation between the two groups.
		Knee strength (handheld dynamometer): • Flexion • Extension	• Significant ↑ in knee flexion from T_1_ to T_2_ and from T_1_ to T_3_ for both groups. • Significant ↑ in knee extension from T_1_ to T_3_ for both groups.

Notes: WBE: water-based exercise group; LBE: land-based exercise group; ↑: increase; ↓: decrease; n = number of participants; M: males; F: females; HRmax: maximal heart rate; HRrest: resting heart rate; HRR: heart rate reserve; VO_2_max: maximal oxygen consumption; VO_2_peak: peak oxygen consumption; VO_2:_ oxygen consumption; BMI: body mass index; RPE: rating of perceived exertion; 6MWT: 6 min walk test; T_1_/T_2_/T_3_: testing points 1/2/3.

## Data Availability

Not applicable.

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
