# Peer review of "The Effectiveness of Water- versus Land-Based Exercise on Specific Measures of Physical Fitness in Healthy Older Adults: An Integrative Review"

_healthcare, 2024, doi:10.3390/healthcare12020221_

Round 1

Reviewer 1 Report

Comments and Suggestions for Authors

Manuscript is interesting. Some minor comments. 
Why did authors classify 50 year old people as older? It seems to me that it’s not the norm. 
Some  citations with errors. 
Why include an article which compared WBE vs LBE + WBE? The aim is LBE vs WBE. 
Please provide more details in quality assessment. Why is your quality assessment better than others?

Please analyze and discuss the particularities of each training and their effects on different outcomes. For example, grip strength, sit up, curl up. Maybe there is some difference between lbe and wbe. 

Reviewer 2 Report

Comments and Suggestions for Authors

Materials and Methods

In a systematic review, the question must be defined in PICOT format.

People under 60 are not considered elderly. This may have affected the number of studies found.

Normally, all systematic reviews are registered on the Prospero website (https://www.crd.york.ac.uk/PROSPERO/), but I did not find this registration in the manuscript.

Results 

What was the criteria for determining the quality of the studies?

Conclusions

"Importantly, considering the individual’s preferences and needs and consulting healthcare professionals is clearly important prior to participating in any new exercise routine. Consequently, future studies involving healthy older individuals of both genders, with higher methodological quality, and, covering a wider range of physical fitness measures are encouraged." This is not conclusion.

Reviewer 3 Report

Comments and Suggestions for Authors

The paper presents an interesting topic, with clinical applicability. 

The abstract provides a clear and concise overview of the study's objective and methods, making it evident that the research aims to compare the effects of water-based versus land-based exercise on physical fitness parameters in healthy older individuals. However, it could be improved by summarizing the main findings more explicitly.

The introduction sets the stage for the study by emphasizing the significance of maintaining an active lifestyle in the aging population. It appropriately highlights the benefits of land-based exercise and the increasing popularity of water-based exercises among older individuals.

The materials and methods section effectively outlines the study design, search strategy, and selection criteria. Please clarify the rationale for selecting the year 1996 as a cut-off date for article inclusion and discuss potential language bias; these aspects would strengthen the section.

Results section is well organized, accompanied by tables and figures relevant to the text. There are some references that are missing (for example in subchapter 1.2. Flexibility line 217, and line 278 in Discussion)

The discussion section, while presenting a general interpretation of the results, lacks a thorough analysis of discrepancies in the findings of the included studies. It could be more expansive and address limitations in greater detail. For example, the impact of gender differences among participants could be explored further. Moreover, a more robust discussion of the implications of the results for clinical practice, future research, and policy would be valuable.

The conclusion summarizes the primary finding, indicating that there is no conclusive evidence favoring either water-based or land-based exercise for improving health-related physical fitness parameters in healthy older adults. However, the conclusion could be more specific by providing clear recommendations for future research or clinical practice.

Comments on the Quality of English Language

 Minor editing of English language required

Round 2

Reviewer 2 Report

Comments and Suggestions for Authors

A systematic review must follow steps and one of them is its registration on the PROSPERO platform and it was admitted that this was not done.

Again, to me, is it necessary include the PICO strategy.

My idea regarding the manuscript is the same: rejection. Justification: every article classified as a systematic review should have been previously registered somewhere for this type of article and this was not done.

Author Response

Dear Reviewer (2),

we acknowledge your time and effort dedicated to providing constructive comments.

As we already mentioned in revision round 1, registering to a perspective registration platform, although desirable, is not mandatory. The work published by Runjic et al. (2019) https://doi.org/10.1016/j.jclinepi.2019.08.010, supports our position.

Moreover, registering in PROSPERO platform may be problematic for users outside UK as “the entire processing time may take up to several months” (Puljak, 2020, https://doi.org/10.1136/bmjebm-2020-111474).

In addition, according to Booth et al. (2012, https://doi.org/10.1186/2046-4053-1-2) “PROPSPERO is restricted to systematic reviews of the effects of interventions and strategies to prevent, diagnose, treat, and monitor health conditions”.

Nevertheless, the aim of our brief systematic review was to analyze research findings related to the effects of different exercise programs on specific physical fitness measures in healthy older individuals.

Regarding the inclusion of PICO framework, again, is it now necessary in the cases where the research question is clear and well-defined, as we believe it happened in our case, while “the effect of using the PICO model as a search strategy tool is still lacking” Eriksen et al. 2018, doi: 10.5195/jmla.2018.345. From what we now, Healthcare Journal does not require this specific framework.

Consequently, we believe that we addressed your concerns raised, aligned with the high standards of Healthcare Journal. We would be grateful for your final evaluation of the manuscript.